

# Contact toxicity of three insecticides for use in tier I pesticide risk assessments with *Megachile rotundata* (Hymenoptera: Megachilidae)

Graham R. Ansell, Andrew J. Frewin, Angela E. Gradish and Cynthia D. Scott-Dupree

School of Environmental Sciences, University of Guelph, Guelph, Ontario, Canada

## ABSTRACT

The current pesticide risk assessment paradigm may not adequately protect solitary bees as it focuses primarily on the honey bee (*Apis mellifera*). The alfalfa leafcutting bee (*Megachile rotundata*) is a potential surrogate species for use in pesticide risk assessment for solitary bees in North America. However, the toxicity of potential toxic reference standards to *M. rotundata* will need to be determined before pesticide risk assessment tests (tier I trials) can be implemented. Therefore, we assessed the acute topical toxicity and generated $LD_{50}$ values for three insecticides: dimethoate (62.08 ng a.i./bee), permethrin (50.01 ng a.i./bee), and imidacloprid (12.82 ng a.i/bee). The variation in the mass of individual bees had a significant but small effect on these toxicity estimates. Overall, the toxicity of these insecticides to *M. rotundata* were within the 10-fold safety factor currently used with *A. mellifera* toxicity estimates from tier I trials to estimate risk to other bee species. Therefore, tier I pesticide risk assessments with solitary bees may not be necessary, and efforts could be directed to developing more realistic, higher-tier pesticide risk assessment trials for solitary bees.

## INTRODUCTION

Bees are important pollinators in natural and agro-ecosystems (*Smagghe & Calderone, 2012*). Solitary bees, which comprise the majority of North American bee species (*Michener, 2007*), are equally or more effective than honey bees (*Apis mellifera*) at pollinating a variety of crops (*Vaughn et al., 2014*) and uncultivated plants. Despite their importance, solitary bees are underrepresented in bee research (*Vaughn et al., 2014*; *EFSA, 2013*).

Solitary bee population declines (*Goulson et al., 2015*; *Koh et al., 2015*; *Potts et al., 2015*; *Leach & Drummond, 2018*) have raised concerns about our ability to predict the potential effect of pesticides on them. Current pesticide risk assessment practices in North America and the European Union focus almost exclusively on *A. mellifera* (*EFSA, 2013*; *EPA, 2014*), and they do not account for the characteristic differences in life history, physiology,

Corresponding author
Graham R. Ansell,
gansell@uoguelph.ca

and behaviour between solitary bees and *A. mellifera* that may result in higher pesticide susceptibility or exposure for solitary bees (*Sgolastra et al., 2018*).

International efforts are underway to develop pesticide risk assessment protocols that will include solitary bees and more accurately assess their routes of exposure and susceptibility to pesticides (*EFSA, 2013*; *EPA, 2014*; *Fischer & Moriarty, 2014*; *OECD, 2017*; *Sgolastra et al., 2018*). As it is unfeasible to produce regulatory guidelines for every species of solitary bee, surrogate test species will need to be selected for different regions (*Vaughn et al., 2014*). The alfalfa leafcutting bee (*Megachile rotundata* Fabricius 1787) has been suggested as a potential surrogate for North American risk assessment for solitary bees, as its biology and behaviour are well understood, and individuals are commercially available in large quantities (*Pitts-Singer & Cane, 2011*).

Regulatory pesticide risk assessment for bees is a three-tiered system involving tier I laboratory, tier II semi-field, and tier III field trials (*OECD, 1998a*; *OECD, 1998b*; *EPA, 2016*). Tier I assessments are used as a screening tool to determine the acute toxicity of pesticides and filter out those that are unlikely to be toxic under proposed use conditions. If certain trigger values indicating potential harm are reached, products are moved to higher levels of testing. The exact protocols currently used are not appropriate for *M. rotundata* due to the behavioural and physical differences between *M. rotundata* and *A. mellifera*. For tier I pesticide risk assessments to be performed with *M. rotundata*, we must develop and use standardized methods that are unique to this species. These will be modified from existing risk assessment methods, but we must standardize the rearing methods, environmental conditions, treatment protocols, and expected toxicity of reference standards for *M. rotundata*. In this paper, we performed a series of toxicity tests to contribute towards the development of a standardized tier I test for *M. rotundata* based on those used by *OECD (1998a)* and similar to *Piccolomini et al. (2018b)*.

The recent development of acute oral toxicity testing methods for *Bombus* spp. (*OECD, 2017*) is an excellent example of what can be done to develop *M. rotundata* topical toxicity tests. The *Bombus* protocols were based on the acute oral toxicity test for *A. mellifera* (*OECD, 1998a*) and modified for use with *Bombus* spp. in North America and the European Union (*OECD, 2017*). Similarly, *A. mellifera* toxicity testing methods (*OECD, 1998b*) can be modified for *M. rotundata,* including modifications to rearing protocols, environmental requirements, and toxic reference standard values that are inherently different between the two species (*OECD, 1998a*; *OECD, 1998b*; *Vaughn et al., 2014*; *Piccolomini et al., 2018b*). Using modified meathods from *OECD (1998a)* allows for robust comparisons of toxicity between species.

First, a reference standard with a known toxic effect is required for tier I pesticide risk assessments: The toxic reference standard is used in parallel with the pesticide under examination to confirm exposure of the test organism during the experiment (*EPA, 2012*). We determined the acute contact toxicity of dimethoate, permethrin, and imidacloprid, potential toxic reference standards for use in tier I trials, to adult female *M. rotundata*. Dimethoate is an organophosphate insecticide currently used as a toxic reference standard in *A. mellifera* risk assessment and suggested as a toxic reference standard for pesticide risk assessment with *Osmia* spp. and *Bombus* spp. (*OECD, 1998b*; *OECD, 2017*; *Uhl et al.,*

*2016*). Permethrin is a commonly used pyrethroid insecticide that has high acute toxicity to *M. rotundata* (Helson, Barber & Kingsbury, 1994; *Piccolomini et al., 2018b*), making it an excellent candidate as a toxic reference standard. Imidacloprid is a neonicotinoid insecticide with systemic activity commonly used as a seed treatment, soil drench, or foliar spray. *EFSA (2012)* suggests that a systemic insecticide be used as one of several toxic reference standards, and current data suggest that imidacloprid is highly toxic to *M. rotundata* (*Scott-Dupree, Conroy & Harris, 2009*).

Second, we assessed the influence of individual *M. rotundata* body mass on the dose received (dose/g bee) from a fixed exposure to pesticide (*Klostermeyer, Stephen & Rasmussen, 1973*; *Peterson & Roitberg, 2006*; *Peterson, Roitberg & Peterson, 2006*). Given that the effective dose of a pesticide received by an individual is inversely proportional to its body mass, larger individuals can be expected to receive a smaller dose of pesticide and therefore generally will be less susceptible to pesticides than smaller individuals. *Uhl et al. (2016)* found body size to have a weak relationship with the insecticide susceptibility of several bee species to topically applied dimethoate. *Devillers et al. (2003)* also reported the susceptibility of *A. mellifera* and non-*Apis* bees to pesticides was inversely proportional to body size across species.

Body mass can also be expected to influence the susceptibility to pesticides between different-sized individuals of the same species. Therefore, in pesticide risk assessment scenarios where all individuals in a treatment receive the same amount of active ingredient, a higher variance in body mass within a population will result in a higher variance in the response to the treatment. *Megachile rotundata* displays a higher variation in body mass between adult individuals than *A. mellifera* (Helson, Barber & Kingsbury, 1994), which may increase the variation of doses received within and between sample test populations. If this variation in individual mass is not measured and included in current pesticide risk assessment methods for bees, it is likely to decrease the precision and consistency of toxicity estimates for *M. rotundata* in tier I trials.

## METHODS

### Test insects

*Megachile rotundata* pre-pupae were purchased from NorthStar Seed Ltd. (Manitoba, Canada) in November the year before the bees were used, and stored at 8 °C. Bees were not used beyond 10 months of purchase to ensure that the adults that emerged were healthy (*Richards, Whitfield & Schaalje, 1987*). Pre-pupae were placed in plastic containers and stored in a temperature-controlled walk-in growth cabinet (Coldstream Products of Canada LTD., Winnipeg, Manitoba, Canada, model WIDF) at 27–30 °C, 60% RH, and 12:12 h light and dark until adult emergence (3–4 wk later). Large holes were cut in the lid of each container and covered on the inside with fine wire mesh to allow for ventilation and prevent the bees from escaping. The holes were covered externally with fine polyester No-see-um fabric netting (Skeeta, Florida, USA) to prevent parasitoid wasps from moving between containers. Containers were checked every other day for parasitoids, and all adult parasitoids and visibly parasitized leaf cells were removed. In the second year, containers

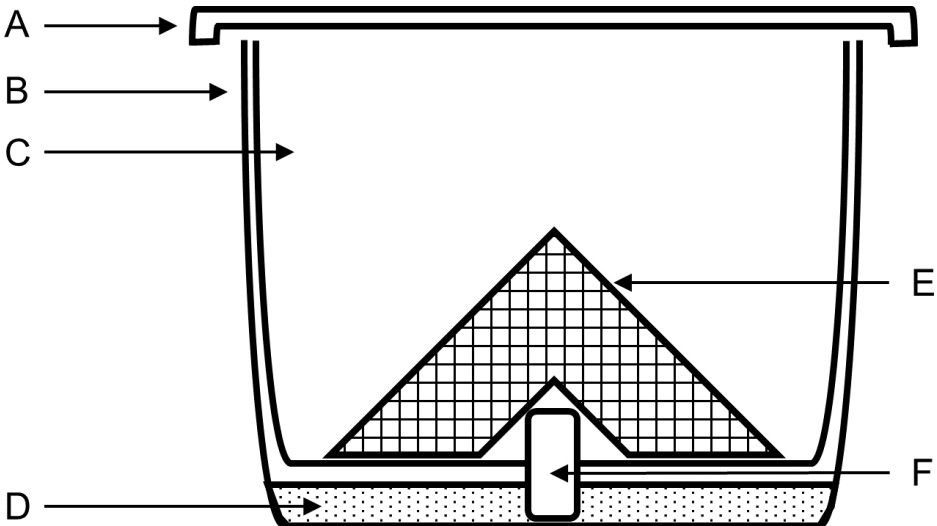

**Figure 1** **Post-treatment container design for topical LD$_{50}$ experiments using *Megachile rotundata*.** (A) screened lid, (B) outer 500-mL clear cup containing sugar solution. (C) inner 500-mL clear cup with hole in center for dental wick. (D) sugar solution. (E) aluminum mesh. (F) dental wick feeder soaking in sugar solution.

were placed in bags made from breathable low tunnel greenhouse screen to further restrict parasitoid mobility. Upon emergence, female bees were stored in groups of 10 in treatment cups (Fig. 1) and fed 20% sucrose solution in a growth cabinet at 25 °C, 50% RH, and 12:12 h light and dark for 3 d to acclimate to experimental conditions. Treatment cups were composed of two 500-mL clear plastic cups, a piece of dental wick soaking in a reservoir of 20% sucrose solution as a food source, and a bent piece of aluminum mesh to provide a surface for the bees to stand on.

## Insecticide treatments

The methods used to generate LD$_{50}$ estimates were adapted from current *A. mellifera* tier I risk assessment (*OECD, 1998b*). Technical grade (90–100% purity) (MilliporeSigma, Ontario, Canada) insecticides were used in the experiments: dimethoate (5 doses were tested, between 10 and 150 ng a.i./bee), permethrin (5 doses between 10 and 90 ng a.i./bee), imidacloprid (5 doses between 0.5 and 40 ng a.i./bee) (Table 1). Insecticides were diluted in acetone. Bees were anesthetized with $CO_2$ from a compressed gas canister for 40 s in groups of 10, and 1 μL of the corresponding insecticide solution was applied to the dorsal thorax of each bee using a micropipette inside a fume hood. Control bees were treated with acetone only. The bees were treated in a randomized complete block design, where blocks were separated across days and each block contained 10 bees of each treatment for a single insecticide (Table 1). Treated bees were placed in new post-treatment containers with fresh 20% sucrose solution and returned to the growth cabinet. Post-treatment containers were arranged randomly on the growth cabinet shelf. Mortality was recorded daily, and the bees were considered dead if they did not respond to a gentle squeeze on the thorax with

**Table 1 Treatments, sample sizes, arithmetic mean mortality, and mean dry mass with standard deviation (SD) of female *Megachile rotundata* 72 h after topical exposure to three insecticides.**

| Insecticide | Treatments (ng a.i./bee) | Sample size | Mean mortality (%) | Mean dry mass (SD) in mg |
|---|---|---|---|---|
| Dimethoate | 0 | 59 | 10.1 | 15.2 (2.42) |
| | 10 | 62 | 9.5 | 15.9 (3.04) |
| | 50 | 63 | 25.3 | 15.9 (3.92) |
| | 100 | 62 | 86.9 | 13.2 (2.63) |
| | 125 | 48 | 95.5 | 13.7 (2.24) |
| | 150 | 54 | 100 | 12.9 (2.28) |
| Permethrin | 0 | 60 | 13.2 | 15.1 (3.15) |
| | 10 | 60 | 45.0 | 13.0 (3.45) |
| | 30 | 60 | 15.0 | 15.8 (3.3) |
| | 50 | 59 | 53.7 | 14.7 (3.37) |
| | 70 | 60 | 82.5 | 13.7 (3.39) |
| | 90 | 60 | 82.5 | 12.9 (2.39) |
| Imidacloprid | 0 | 70 | 10 | 15.0 (2.12) |
| | 0.5 | 69 | 23.2 | 12.9 (2.28) |
| | 10 | 60 | 35.0 | 12.6 (2.29) |
| | 20 | 70 | 71.4 | 12.5 (1.90) |
| | 30 | 70 | 84.3 | 12.87 (1.92) |
| | 40 | 70 | 90.0 | 12.4 (1.88) |

forceps. After 3 d, all bees were placed in the freezer for at least 1 d, rinsed with de-ionized water, desiccated thoroughly in a drying oven at 47 °C, and weighed.

## Statistical analyses

All significance values were tested at $\alpha = 0.05$. Data were analyzed using a generalized linear model in R with the glm function using a binomial distribution and probit transformation (*R Core Team, 2017*). Individual bee mass was used as an explanatory variable, and each bee was treated as an experimental unit, as all bees were maintained under virtually identical conditions. The significance of each explanatory variable was tested with a Wald test (*Agresti, 1990*). Control mortality was corrected for using the Henderson-Tilton equation (*Henderson & Tilton, 1955*). The data were also analyzed without mass as an explanatory variable to assess the magnitude of the effect of individual bee mass on toxicity estimates. Model fit was approximated by calculating the pseudo $R^2$ using Eq. (1) (*Alain et al., 2009*).

$$\text{pseudo } R^2 = 100 \times \frac{\text{Null deviance} - \text{residual deviance}}{\text{null deviance}}. \tag{1}$$

Hazard quotients were calculated using the highest field rate (Eq. (2)) of one formulated product of each insecticide and the $LD_{50}$ generated in this study (*EFSA, 2013*). Hazard quotients are required in the European Union to determine if a hazard trigger value is reached (*EFSA, 2013*), but a risk quotient would normally be used for North American bee pesticide risk assessment (*EPA, 2012*). A risk quotient incorporates the likelihood of

**Table 2  Topical toxicity of dimethoate, permethrin, and imidacloprid to *Megachile rotundata* females.**  The toxicity of each insecticide was determined with statistical models that included or did not include the body mass of individual bees as a covariate. Wald test $X^2$ statistics are provided for both the effect of treatment and the effect of mass, where a large test statistic and significant P value indicate a significant effect on the model. Model parameters were generated using probit-transformed mortality data and untransformed dose data.

| Insecticide | Individual mass included as covariate | LD$_{50}$ ($\pm$ 95% CI) (ng a.i./bee) | Slope at the LD$_{50}$ | Wald X$^2$ treatment | Wald X$^2$ P treatment | Wald X$^2$ mass | Wald X$^2$ P mass | Pseudo R$^2$ |
|---|---|---|---|---|---|---|---|---|
| Dimethoate | No | 69.84 (7.97) | 2.62 e$^{-2}$ | 256.03 | <0.0001 | NA | NA | 53.57 |
| | Yes | 61.74 (8.56) | 2.85 e$^{-2}$ | 256.03 | <0.0001 | 98.14 | <0.0001 | 74.11 |
| Permethrin | No | 53.49 (8.25) | 2.13 e$^{-2}$ | 59.142 | <0.0001 | NA | NA | 17.88 |
| | Yes | 44.89 (8.17) | 2.66 e$^{-2}$ | 29.142 | <0.0001 | 92.69 | <0.0001 | 45.84 |
| Imidacloprid | No | 17.36 (2.73) | 6.13 e$^{-2}$ | 114.16 | <0.0001 | NA | NA | 28.75 |
| | Yes | 12.90 (3.05) | 6.38 e$^{-2}$ | 282.87 | <0.0001 | 57.50 | <0.0001 | 43.24 |

exposure to the pesticide and produces a more accurate assessment of risk than the more conservative hazard quotient. Unfortunately, there are not enough data on *M. rotundata* pesticide exposure to accurately generate risk quotients at this time (*EPA, 2012*).

$$\text{Hazard Quotient} = \frac{\text{application rate(ga.i./ha)}}{\text{LD}_{50}}. \tag{2}$$

The LD$_{50}$ values were compared with previous values calculated for *A. mellifera* with and without accounting for the mean mass of an *A. mellifera* adult worker and the mean mass of the *M. rotundata* females in this study. Further, a 10-fold safety factor was calculated from these *A. mellifera* values to address if the current tier I pesticide risk assessment methods are protective of *M. rotundata* when species body mass is accounted for.

## RESULTS

Imidacloprid was more toxic than permethrin, which was slightly more toxic than dimethoate (Table 1). The slope of the dose response curve across all methods of bee mass incorporation was steepest for imidacloprid, followed by dimethoate, and then permethrin (Table 2, Fig. 2). When the mean mass of individual adults of each species (*M. rotundata* and *A. mellifera*) was not taken into account, imidacloprid and dimethoate were more toxic to *M. rotundata* than *A. mellifera* (Table 3). However, all three insecticides were less toxic to *M. rotundata* than *A. mellifera* when the mean mass of individual adults was accounted for (Table 3). The mean dry mass of all bees in these trials was 14.4 mg with a standard deviation of 3.06 mg.

All surviving bees treated with imidacloprid at all doses exhibited various degrees of rigid paralysis (*Sharf, 2008*) for the duration of the experiment, indicating that imidacloprid affects bees for more than 72 h. It is thus unclear how many bees would fully recover, remain moribund, or die from imidacloprid exposure. Paralyzed bees were characterized as having their legs and abdomen uncurled with their wings together behind the thorax in a posture similar to a resting bee, but nearly unable to move. Paralyzed bees were observed both ventral side down and ventral side up, and responded to forceps stimulation with varying

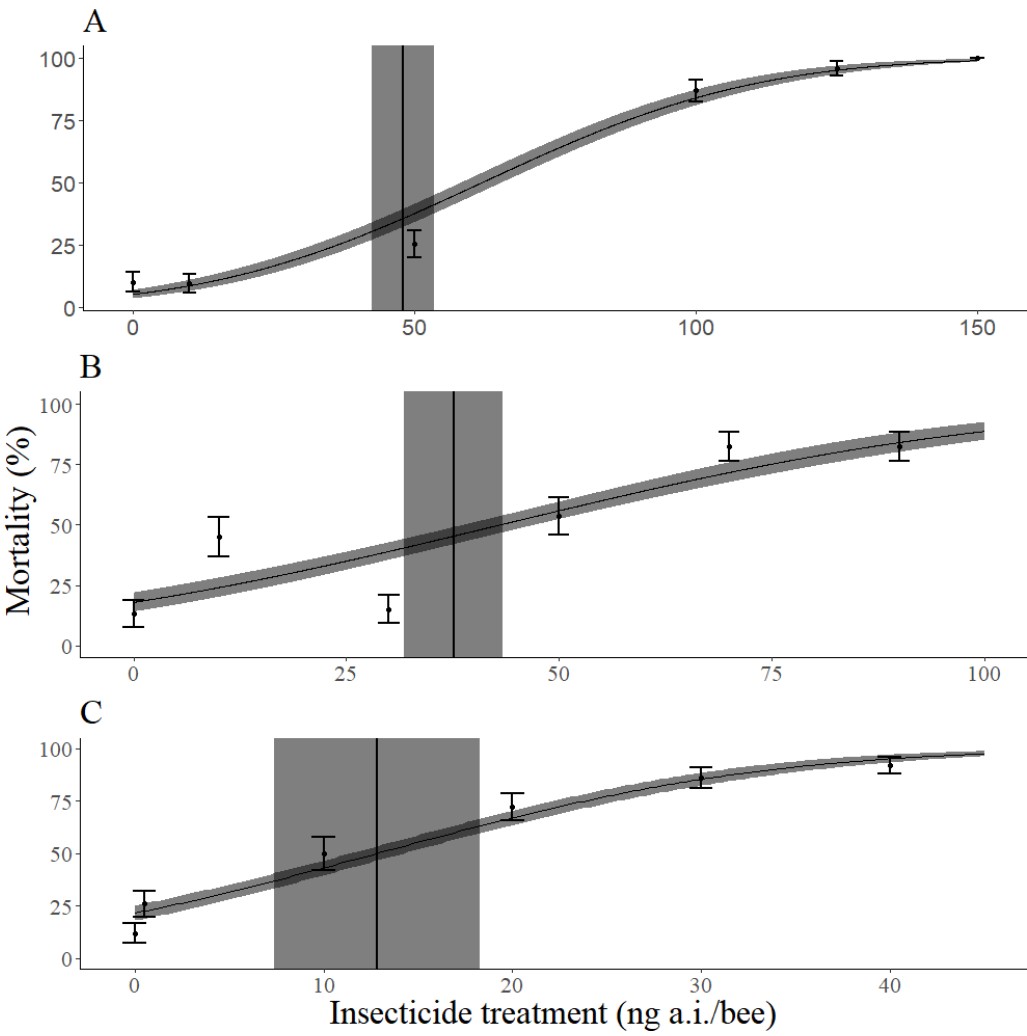

**Figure 2  Dose response curves for mortality 72 h after topical exposure of female *Megachile rodundata* to dimethoate (A), permethrin (B), or imidacloprid (C).** These models did not include the body mass of individual bees as a covariate. $LD_{50}$ values (ng a.i./bee) and 95% confidence intervals are represented by vertical black lines and gray rectangles, respectively. The standard error of predicted values is represented by the gray ribbon around the dose response curve.

degrees of leg and antennal twitching. The degree of paralysis seemed to be dose-dependent, where bees that were treated with higher doses were less able to move than bees treated with lower doses. Bees treated with dimethoate exhibited varying degrees of jerky movements such as abdominal spasms, uncoordinated locomotion, and difficulty righting (*Williamson et al., 2013*) in the first day, and the bees either died or recovered after 72 h. Bees treated with permethrin did not exhibit any sublethal toxicity symptoms.

Mean bee mass was a significant factor in the models (Table 2). However, the actual $LD_{50}$ values generated when incorporating and not incorporating individual mass within each insecticide were not different, as in all cases the confidence values overlapped between

**Table 3 Toxicity of dimethoate, permethrin, and imidacloprid to *Megachile rotundata* 72 h after topical exposure.** $LD_{50}$ values for *M. rotundata* from this study are presented along with the relative toxicity to *Apis mellifera* and previous estimates for *M. rotundata*. $LD_{50}$ values incorporating mean body mass for each species are estimated using the mean mass of individual *M. rotundata* adults in this study and the mean mass of individual *A. mellifera* adults.

| Pesticide | $LD_{50}$ (± 95% CI) (ng a.i./bee) | Toxicity ratio to honey bees (per bee) | Toxicity ratio to honey bees (per mean body mass) |
|---|---|---|---|
| Dimethoate | 62.08 (7.05) | 1: 3.75[a] | 1: 0.4d[c] |
| Permethrin | 50.01 (8.17) | 1: 1.46[b] | 1: 0.16[b,c] |
| Imidacloprid | 12.82 (2.78) | 1: 6.13[b] | 1: 0.62[b,c] |

Notes.
[a] *OECD (1998b)*.
[b] *Sanchez-Bayo & Goka (2014)*.
[c] *EPA (2012)*.

**Table 4 Hazard quotients of imidacloprid, dimethoate, and permethrin to *Megachile rotundata*.** Hazard quotients were calculated using acute topical toxicity values and the highest legal application rate from a formulated product containing each insecticide that is currently registered in Canada and the United States. The trigger value for solitary bees suggested by *EFSA (2013)* is a hazard quotient of 8–16 g a.i./ha/g a.i./bee.

| Insecticide | $LD_{50}$ (± 95% CI) (ng a.i./bee) | Highest application rate (g a.i./ha) | Hazard quotient $\left(\frac{\text{g a.i./ha}}{\mu\text{g a.i./bee}}\right)$ |
|---|---|---|---|
| Dimethoate | 62.08 (7.05) | 1104[a] | 17783.51 |
| Permethrin | 50.01 (8.17) | 70[b] | 1399.72 |
| Imidacloprid | 12.82 (2.78) | 48[c] | 3744.15 |

Notes.
[a] Lagon® 480 E, Loveland Canada Products Inc., Dorchester, Ontario, Canada.
[b] Ambush® 500 EC, AMVAC Chemical Corporation, Newport Beach, California, USA.
[c] Admire® 240, Bayer CropScience Inc., Calgary, Alberta, Canada.

estimates (Table 2). Models that did not include mass had higher fit statistics than models that did include bee mass within each insecticide (Table 2).

The hazard quotient value for dimethoate surpassed the proposed trigger value for solitary bees (*EFSA, 2013*), while the hazard quotients for permethrin and imidacloprid did not (Table 4).

## DISCUSSION

Thus far, our study is the only assessment of the toxicity of permethrin, dimethoate, or imidacloprid for the purposes of tier I *M. rotundata* pesticide risk assessment method development. Based on our results, imidacloprid is a poor choice as a toxic reference standard because the paralysis it induced for the duration of this experiment made it difficult to assess mortality. Additionally, imidacloprid will likely continue to have a negative effect on survival beyond the short duration of a tier I risk assessment (usually 48 or 72 h) because none of the bees treated with imidacloprid fully recovered in that time

frame. In contrast, the observable sublethal effects of dimethoate and permethrin ceased by the end of the experiment, indicating that all individuals had either died or recovered. The cessation of toxic effects within a short time is beneficial when designing short-term trials as the onset and conclusion of effects are clear and occur within the test period. Using dimethoate as a toxic reference standard for tier I *M. rotundata* risk assessment would maintain continuity with other bee pesticide risk assessment methods, all of which use dimethoate, (*OECD, 1998b*; *OECD, 2017*; *Knäbe et al., 2017*). However, ring testing will be required to implement any of these insecticides as toxic reference standards for *M. rotundata* pesticide risk assessment.

Our results suggest that incorporating individual mass in a dose response model will not improve the precision of the $LD_{50}$ or model fit for acute topical application of insecticides to *M. rotundata*. Similarly, *Helson, Barber & Kingsbury (1994)* found that incorporating the mass of individuals in their models did not change $LD_{50}$ estimates when assessing the effect of acute topical applications of six pesticides on four bee species (*M. rotundata*, *A. mellifera*, *Andrena erythronii*, and *Bombus terricola*). Although we found that mass had a significant effect on our models, there was not a significant change in toxicity estimates, and model fit decreased when individual mass was incorporated. Therefore, we conclude that incorporating individual mass when calculating $LD_{50}$ estimates for the purposes of pesticide risk assessment with *M. rotundata* is not necessary.

However, we still recommend reporting the mean body mass of test populations in solitary bee pesticide risk assessment. The mean body mass of *M. rotundata* is affected by latitude (*Pankiw, Lieverse & Siemens, 2012*), production protocols (*Pitts-Singer & Cane, 2011*), environmental conditions, and food quality and quantity (*Klostermeyer, Stephen & Rasmussen, 1973*; *Rothschild, 1979*). As risk assessments are based on data compiled from multiple experiments using sample populations from different locations, it would be prudent to report the mean body mass of the bees in these various experimental groups to account for some of the differences in estimates generated from different sample populations from different locations. This may help contextualize results across studies, and is a relatively simple endpoint to measure.

The hazard quotient for all three insecticides we studied (Table 4) vastly exceeded the trigger values for solitary bees (8–16 $\frac{\text{ga.i./ha}}{\mu\text{ga.i./bee}}$) proposed by *EFSA (2013)*. The trigger values proposed by EFSA do not account for exposure and are more conservative than the risk quotient trigger values usually used in North America (*EPA, 2012*). However, without the *M. rotundata* life history data required to generate a risk quotient, the hazard quotient generated here is the most accurate estimate of potential risk to *M. rotundata*. Therefore, our data support that the current safety factors used in North America to protect solitary bees in the field should be re-examined.

Our results were similar to those found in other recent topical toxicity studies with *M. rotundata*, although the variation in methods between those studies and ours prevent direct comparison. Our 72 h $LD_{50}$ for permethrin (53.49 ng a.i./bee) was similar to the 24 h $LD_{50}$ of 57 ng a.i./bee reported by *Piccolomini et al. (2018b)*. It was also similar to the 48 h $LD_{50}$ of 18 ng a.i./bee reported by Helson, Barber & Kingsbury (1994), although they kept their bees in post-treatment containers at 16 C˚which is almost 10 C˚cooler than other

experiments. Our 72 hr $LD_{50}$ for imidacloprid (17.36 ng a.i./bee) was within two–fold of the 48 h $LD_{50}$ generated by *Hayward et al. (2019)* of 10 ng/bee.

Our results suggest that while safety factors for solitary bees may need to be re-examined, tier I solitary bee pesticide risk assessment may not be necessary within the current framework. The toxicity of all 3 insecticides in our study were within the 10-fold assessment factor (1/10 the $LD_{50}$ for *A. mellifera*) proposed by *EFSA* when compared to *A. mellifera* toxicity estimates from the literature. This is consistent with the current literature where in nearly all cases solitary bees are protected by the assessment factor: *Arena & Sgolastra (2014)* reported that 95% of bee species studied in 44 laboratory trials were protected within the 10-fold assessment factor for 6 pesticides, Helson, Barber & Kingsbury (1994) found that 3 bee species, including *M. rotundata*, were protected, and *Uhl et al. (2016)* found the 10-fold assessment factor protective for 5 European bee species exposed to dimethoate. *Piccolomini et al. (2018b)* also reported similar $LD_{50}$ values for 3 pyrethroids (including permethrin) for *M. rotundata* and *A. mellifera*, and it was further demonstrated that the pyrethroid etofenprox was also not acutely toxic to *M. rotundata* in field trials with commercial *M. rotundata* nesting units (*Piccolomini et al., 2018a*). Furthermore, in our study *M. rotundata* was less susceptible than *A. mellifera* to all three of the insecticides when body mass was accounted for, similar to the results reported by Helson, Barber & Kingsbury (1994).

*Hayward et al. (2019)* also found that *M. rotundata* was within the 10-fold assessment factor for imidacloprid, but was 2,500 fold more sensitive to thiacloprid and 170 fold more sensitive to flupyradifurone than *A. mellifera*. The extreme sensitivity exhibited by *M. rotundata* to thiacloprid and flupyradifurone is due to a lack of P450 enzymes in the CYP9Q subfamily that are present other managed bees (*Hayward et al., 2019*). Therefore, it appears that in nearly all cases the threshold values that are protective to *A. mellifera* via topical exposure in a laboratory environment are likely to be protective of *M. rotundata* based on the 10-fold assessment factor. Instead of developing costly protocols to address the few cases where the assessment factor is not protective, alternative approaches may be more effective. Perhaps an additional assessment factor may be developed for species lacking certain P450 enzymes, or insecticides that are detoxified by those enzymes could be passed on to higher-tier tests automatically. Tier I tests may even be developed for these fringe cases alone. We will not know the best course of action until we understand more about solitary bees' responses to insecticides.

Despite the protectiveness of the 10-fold assessment factor in tier I trials, *A. mellifera* is still unlikely to be a suitable surrogate for all bee species at the tier II and III scale. Laboratory exposure experiments do not reflect many of the key differences in life history, behavior, and sociality between *M. rotundata* and *A. mellifera* (*EFSA, 2013*; *Sgolastra et al., 2018*). For example, under more realistic conditions, solitary bee larvae may experience higher pesticide exposure than *A. mellifera* larvae via contact with residues on leaf cuttings or soil used to construct their cells, and as adults with smaller foraging ranges when in close proximity to agriculture (*Vaughn et al., 2014*). Additionally, the death of a single female solitary bee has a larger impact on the population's reproductive potential than the death of a single *A. mellifera* worker. These behavioural and life history differences are more

likely to result in differences in susceptibility and exposure of solitary bees and *A. mellifera* at higher tiers of risk assessment that cannot be extrapolated from tier I results.

## CONCLUSION

Our data suggest that dimethoate or permethrin may be suitable as a toxic reference standard for tier I pesticide risk assessment for *M. rotundata*. However, thorough replication and method development will be required before either insecticide can be incorporated into the risk assessment process. Although we did not find that individual bee mass influences $LD_{50}$ estimates for *M. rotundata*, we recommend reporting the mean mass of sample populations as they may vary between experiments and could affect toxicity estimates between studies. Finally, our results suggest that it may not be necessary to develop topical tier I pesticide risk assessment using *M. rotundata*, as the $LD_{50}$ values that we generated were within a 10–fold assessment factor of previously generated *A. mellifera* $LD_{50}$ values as suggested by *EFSA (2013)*.

### Funding
This study was funded by a Mitacs-Elevate grant in partnership with Syngenta Canada awarded to Andrew J. Frewin. The funders had no role in study design, data collection and analysis, decision to publish, or preparation of the manuscript.

### Grant Disclosures
The following grant information was disclosed by the authors:
Mitacs-Elevate grant in partnership with Syngenta Canada awarded to Andrew J. Frewin.

### Competing Interests
The authors declare there are no competing interests.

### Author Contributions
- Graham R. Ansell conceived and designed the experiments, performed the experiments, analyzed the data, prepared figures and/or tables, authored or reviewed drafts of the paper, and approved the final draft.
- Andrew J. Frewin conceived and designed the experiments, performed the experiments, authored or reviewed drafts of the paper, and approved the final draft.
- Angela E. Gradish and Cynthia D. Scott-Dupree conceived and designed the experiments, authored or reviewed drafts of the paper, and approved the final draft.

### Data Availability
All the raw mortality and individual bee mass data are available in the Supplemental Files.

## Supplemental Information

Supplemental information for this article can be found online at http://dx.doi.org/10.7717/peerj.10744#supplemental-information.

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
