# Peer review of "Contact toxicity of three insecticides for use in tier I pesticide risk assessments with Megachile rotundata (Hymenoptera: Megachilidae)"

_PeerJ, doi:10.7717/peerj.10744_

## Round 0.1 · original submission · Major Revisions

Dear Dr. Ansell and colleagues:

Thanks for submitting your manuscript to PeerJ. I have now received two independent reviews of your work, and as you will see, the reviewers raised some concerns about the research. Despite this, this the reviewers are optimistic about your work and the potential impact it will have on research studying pesticide risk assessment. Thus, I encourage you to revise your manuscript, accordingly, taking into account all of the concerns raised by both reviewers.

While the concerns of the reviewers are relatively minor, this is a major revision to ensure that the original reviewers have a chance to evaluate your responses to their concerns.

Importantly, please ensure your Materials and Methods are clearly stated. The methods should be clear, concise and repeatable. Please ensure this, and make sure all relevant information and references are provided. Also, please reframe your major conclusion based on the concern of Reviewer 2.

There are many minor suggestions to improve the manuscript (typos, nuances, etc.).

I agree with many of the concerns of the reviewers, and thus feel that their suggestions should be adequately addressed before moving forward.

I look forward to seeing your revision, and thanks again for submitting your work to PeerJ.

Good luck with your revision,

-joe

·

Basic reporting

See General comments for the author

Experimental design

See General comments for the author

Validity of the findings

See General comments for the author

Additional comments

The manuscript reports LD50 results for three insecticides on the alfalfa leafcutting bee, Megachile rotundata. The manuscript provides results of interest to better understand risks to that species as well as other solitary bee species. The paper is generally well written and methodologically sound.

Comments:
L1-2. The title is trying to capture something the paper is not. These aren’t preliminary tests nor are they potential insecticides. The authors are simply reporting LD50 values for three insecticides––and that’s good enough! Issues of methods development and assessment and benchmark data for other insecticides can be discussed in the text. It does not need to be captured in the title.
L2. Should it be “(Hymenoptera: Megachilidae)” instead?
L18. Insert “Therefore,” before “we…”
L30-31. Change “Vaugh” to “Vaughn”.
L44. Consider not using the abbreviation “ALB”. Use “M. rotundata” or “alfalfa leafcutting bee” instead.
L48. Change to “Regulatory pesticide risk assessment for bees is a…”
L53. The authors state, “…we must develop and use standardized methods that are unique to this species.” Why? Provide a rationale for this, especially given that the authors derive a LD50 for permethrin of 50 ng/bee, which is nearly very similar to the LD50 of 57 ng/bee derived by Piccolomini et al. (2018) using very different methods and statistical analyses. Also, Piccolomini et al. (2018) essentially used the same EPA SOP as that for honey bees. So, again, what is the rationale for the authors’ statement?
L70. Change “suggests” to “suggest”.
L100. Provide more information on the growth chamber (e.g., brand, model, dimensions, location of manufacturer).
L113-127. This section could be improved with a clearer description of several aspects of the methods. It is unclear how many bees were used per dose per experiment. Also, list the 5 doses, not just the range between the lowest and highest. The authors state, “The bees were treated in a randomized complete block design, where blocks were separated across days.” What do the blocks represent? All doses of a single insecticide? A group of bees treated with one dose of an insecticide? How many experimental replications were conducted for each insecticide at each dose?
L119. What was the ppm of the CO2 used?
L143. Change “vale” to “value”.
L147. Change “is” to “are”.
L171. Change “appeared” to “seemed”.
L190-191. There needs to be a “because” after this statement. Provide an explanation.
L204. “rotundata” needs to be italicized.
L221. How is the hazard quotient dimensionless when the units do not cancel?
L245. Change “honeybees” to “honey bees”.
L253-262. Can this section be deleted? What does it add that hasn’t already been stated?
Discussion. The authors might strengthen their arguments by discussing complementary information from Piccolomini et al. (2018) (e.g., very similar LD50 values for permethrin and discussion of Apis mellifera in fourth and last paragraph in their discussion section). Also, another paper addresses the authors’ comments about tier 1 vs. tier 3 risk assessment using M. rotundata. See Piccolomini et al. 2018. The effects of an ultra-low-volume application of etofenprox for mosquito management on Megachile rotundata (Hymenoptera: Megachilidae) larvae and adults in an agricultural setting. Journal of Economic Entomology. 111:33-38.
Figure 2. A mean and the two error bars for the control for imidacloprid indicate greater than 20% mortality. A standard practice in insect toxicology (and broader than insect toxicology) is to discard trials in which 20% or more control individuals die (Yu, S.J. 2008. The toxicology and biochemistry of insecticides. CRC Press Taylor Francis Group, Boca Raton, FL, USA).
Table 4. Hazard quotients are not dimensionless in this case.

Reviewer 2 ·

Basic reporting

However, the manuscript is principally well referenced, but to my opinion the authors omitted to cite a key paper recently published by Hayward et al. (2019; Nature Ecology & Evolution 3, 1521-24), and showing that M. rotundata is up to 2,500-fold more sensitive to certain insecticides (e.g. thiacloprid) commonly considered selective and honey bee safe.

Experimental design

The paper is generally well-written, and the methods described in enough detail. However, it would be beneficial to mention under M&M the number of replicates tested and how often the bioassays were repeated.

Validity of the findings

I have a fundamental problem with the conclusion drawn by the authors. Please refer to the general comments for the author

Additional comments

Ansell et al. investigated the contact toxicity of three different insecticides with different modes of action against alfalfa leafcutter bee, Megachile rotundata, an important managed pollinator for alfalfa seed production. In contrast to honey bees it is a solitary bee species not forming large colonies. All three insecticides – dimethoate, permethrin and thiamethoxam – were highly toxic as previously shown for other bee pollinators. The employed method was adapted from current honey bee tier I risk assessment according to OECD. The paper is generally well-written, and the methods described in enough detail. However, it would be beneficial to mention under M&M the number of replicates tested and how often the bioassays were repeated. However, the manuscript is principally well referenced, but to my opinion the authors omitted to cite a key paper recently published by Hayward et al. (2019; Nature Ecology & Evolution 3, 1521-24), and showing that M. rotundata is up to 2,500-fold more sensitive to certain insecticides (e.g. thiacloprid) commonly considered selective and honey bee safe.

Therefore, I have a major concern and a fundamental problem regarding the conclusion drawn by the authors: “ Finally, our results suggest that it may not be necessary to develop topical tier I pesticide risk assessment using ALB, as the…”. That said, I think the authors´ conclusion is only correct considering the three toxic insecticides they tested, but to generally conclude tier I pesticide risk assessment with solitary bees may not be necessary seem not tenable, at least in the light of the current scientific knowledge. The authors should consider refining their conclusions or at least discussing their findings in the light of the most recent literature, almost suggesting the opposite in terms of solitary bee tier I pesticide risk assessment.

---

## Round 0.2 · Minor Revisions

Dear Dr. Ansell and colleagues:

Thanks for revising your manuscript. The reviewers are very satisfied with your revision (as am I). Great! However, there are a few minor issues to entertain. Please address these ASAP so we may move towards acceptance of your work.

Best,

-joe

·

Basic reporting

See general comments to the author.

Experimental design

See general comments to the author.

Validity of the findings

See general comments to the author.

Additional comments

The revision is acceptable. I have two minor points:
1) "sp." and "spp." should not be italicized.
2) In my original review, I asked "How is the hazard quotient dimensionless when the units do not cancel?" The authors responded by stating, "Hazard quotients are not usually presented with dimensions and we have presented our data following this trend (e.g., see EFSA, 2012). HQ is also referred to without units in the Europe EU directive 91/414 in reference to honey bee pesticide risk assessment." It's not sufficient justification for the authors of this paper to state that "hazard quotients are not usually presented with dimensions and we have presented our data following this trend." Hazard quotients, risk quotients, and hazard indices are dimensionless because the units cancel. If the units don't cancel, presenting these ratios as dimensionless doesn't mean anything. If EFSA (2012) provides a rationale for why these are presented as dimensionless even when the units don't cancel, then the authors need to explain that. Is it possible that the EFSA is in error? The authors need to thoroughly investigate this.

Reviewer 2 ·

Basic reporting

All points raised in my review were adequately addressed

Experimental design

All points raised in my review were adequately addressed

Validity of the findings

All points raised in my review were adequately addressed

Additional comments

All points raised in my review were adequately addressed

---

## Round 0.3 · accepted · Accept

Dear Dr. Ansell and colleagues:

Thanks for revising your manuscript based on the concerns raised by the reviewer. I now believe that your manuscript is suitable for publication. Congratulations! I look forward to seeing this work in print, and I anticipate it being an important resource for groups studying pesticide risk assessment. Thanks again for choosing PeerJ to publish such important work.

Best,

-joe